# Phenotypic Characterization and Heterogeneity among Modern Clinical Isolates of *Acinetobacter baumannii*

Adam Valcek,[a,b] Chantal Philippe,[c] Clémence Whiteway,[a,b] Etienne Robino,[a,b] Kristina Nesporova,[a,b] Mona Bové,[d] Tom Coenye,[d] Tim De Pooter,[e,f] Wouter De Coster,[g,h] Mojca Strazisar,[e,f] Charles Van der Henst[a,b]

aMicrobial Resistance and Drug Discovery, VIB-VUB Center for Structural Biology, VIB, Flanders Institute for Biotechnology, Brussels, Belgium
bStructural Biology Brussels, Vrije Universiteit Brussel (VUB), Brussels, Belgium
cResearch Unit in the Biology of Microorganisms (URBM), NARILIS, University of Namur (UNamur), Namur, Belgium
dLaboratory of Pharmaceutical Microbiology, Ghent University, Ghent, Belgium
eNeuromics Support Facility, VIB Center for Molecular Neurology, VIB, Antwerp, Belgium
fDepartment of Biomedical Sciences, University of Antwerp, Antwerp, Belgium
gApplied and Translational Neurogenomics Group, VIB Center for Molecular Neurology, VIB, Antwerp, Belgium
hApplied and Translational Neurogenomics Group, Department of Biomedical Sciences, University of Antwerp, Antwerp, Belgium

Adam Valcek and Chantal Philippe contributed equally to the study. Author order was determined on the basis of their contributions.

**ABSTRACT** *Acinetobacter baumannii* is an opportunistic pathogenic bacterium prioritized by WHO and CDC because of its increasing antibiotic resistance. Heterogeneity among strains represents the hallmark of *A. baumannii* bacteria. We wondered to what extent extensively used strains, so-called reference strains, reflect the dynamic nature and intrinsic heterogeneity of these bacteria. We analyzed multiple phenotypic traits of 43 nonredundant, modern, and multidrug-resistant, extensively drug-resistant, and pandrug-resistant clinical isolates and broadly used strains of *A. baumannii*. Comparison of these isolates at the genetic and phenotypic levels confirmed a high degree of heterogeneity. Importantly, we observed that a significant portion of modern clinical isolates strongly differs from several historically established strains in the light of colony morphology, cellular density, capsule production, natural transformability, and *in vivo* virulence. The significant differences between modern clinical isolates of *A. baumannii* and established strains could hamper the study of *A. baumannii*, especially concerning its virulence and resistance mechanisms. Hence, we propose a variable collection of modern clinical isolates that are characterized at the genetic and phenotypic levels, covering a wide range of the phenotypic spectrum, with six different macrocolony type groups, from avirulent to hypervirulent phenotypes, and with naturally noncapsulated to hypermucoid strains, with intermediate phenotypes as well. Strain-specific mechanistic observations remain interesting *per se*, and established "reference" strains have undoubtedly been shown to be very useful to study basic mechanisms of *A. baumannii* biology. However, any study based on a specific strain of *A. baumannii* should be compared to modern and clinically relevant isolates.

**IMPORTANCE** *Acinetobacter baumannii* is a bacterium prioritized by the CDC and WHO because of its increasing antibiotic resistance, leading to treatment failures. The hallmark of this pathogen is the high heterogeneity observed among isolates, due to a very dynamic genome. In this context, we tested if a subset of broadly used isolates, considered "reference" strains, was reflecting the genetic and phenotypic diversity found among currently circulating clinical isolates. We observed that the so-called reference strains do not cover the whole diversity of the modern clinical isolates. While formerly established strains successfully generated a strong base of knowledge in the *A. baumannii* field and beyond, our study shows that a rational choice of strain, related to a specific biological question, should be taken into consideration. Any data

Address correspondence to Charles Van der Henst, charles.vanderhenst@vub.vib.be.

The authors declare no conflict of interest.

10.1128/spectrum.03061-22 **1**

obtained with historically established strains should also be compared to modern and clinically relevant isolates, especially concerning drug screening, resistance, and virulence contexts.

**KEYWORDS** *Acinetobacter baumannii*, macrocolonies, transmission electron microscopy, bacterial capsule, capsular polysaccharide, genotypes, Gram-negative bacteria, phenotypes, virulence factors

Antibiotic overuse, along with a decrease in new therapeutics, represents a major challenge for human health (1, 2). Multidrug-resistant (MDR) bacteria are increasingly isolated around the world, leaving physicians facing more and more a lack of therapeutic options (3). As a direct consequence, patients are currently dying from previously treatable diseases. In this context, WHO and CDC prioritized problematic bacterial pathogens for which antibiotic resistance significantly impacts human health (4, 5). *Acinetobacter baumannii* (6), a member of the ESKAPE (*Enterococcus faecium*, *Staphylococcus aureus*, *Klebsiella pneumoniae*, *Acinetobacter baumannii*, *Pseudomonas aeruginosa*, and *Enterobacter* species) multidrug-resistant and most problematic nosocomial pathogens (7), was designated a top priority and critical agent for which therapeutic alternatives are urgently required (5).

*A. baumannii* is a Gram-negative opportunistic bacterial pathogen that thrives in hospital settings, especially in intensive care units where weakened patients are treated (6). *A. baumannii* is capable of forming biofilms, which are communities of microorganisms which adhere to a/biotic surfaces embedded in the extracellular matrix. Biofilms contribute to chronic and persistent infections, resistance to antibiotics, disinfectants, and host immune defense resistance, and survival under unfavorable conditions such as hospital settings (8). The major role in biofilm formation and adhesion to host cells in *A. baumannii* is attributed to biofilm-associated protein (BAP) (9).

Besides the bacterium's intrinsic and acquired antibiotic resistance, the resistance of *A. baumannii* to desiccation and disinfectants renders any decontamination strategy a real challenge (10). One key aspect sustaining the rapid spread of antibiotic resistance among *A. baumannii* isolates is their capacity for natural transformation (11). *A. baumannii* bacteria undergo horizontal gene transfer during which exogenous DNA is taken up and integrated within the bacterial genome (12). As a direct consequence, rapidly evolving *A. baumannii* bacteria show a dynamic genome, with an estimated core (conserved) genome of only 16.5%, while 25% of the genome is unique to each strain, having no counterpart in any other *A. baumannii* genome (13). However, despite their clinical relevance, *A. baumannii* bacteria remain poorly understood. Especially, their virulence and non-antibiotic-associated resistance still need to be better characterized (14). One possible explanation for this gap of knowledge is the heterogeneity among *A. baumannii* isolates, rendering the multiple approaches to typing this bacterial species difficult, with, e.g., the huge diversity of the polysaccharide capsules and the outer core of the lipooligosaccharide (LOS) (15), in addition to the two multilocus sequence typing (MLST) schemes (16).

LOS is present in *A. baumannii* instead of the common lipopolysaccharide (LPS). Unlike LPS, the LOS lacks the O antigen and is composed of lipid A with variable amounts of inner and outer core sugars (17). The outer sugars of LOS show diversity across strains, dependent on glycosyltransferases and nucleotide-sugar biosynthesis enzymes encoded in a highly variable lipooligosaccharide outer core locus (OCL) (18). Chromosomally encoded LOS is a major virulence component in Gram-negative bacteria (19). To date, 14 variants (differing in the presence or absence of genes encoding multiple glycosyltransferases and other enzymes) were described in *A. baumannii*. OCL types can be divided into two groups (A and B) based on the presence of *pda1* and *pda2* genes (17). The OCL consists of genes involved in the synthesis, assembly, and export of complex oligosaccharides that are then linked to lipid A to form the LOS (20). The bacterial capsule consists of a polysaccharide layer deposited as the outermost surface-exposed leaflet, impacting virulence and antibiotic resistance, as well as non-antibiotic-

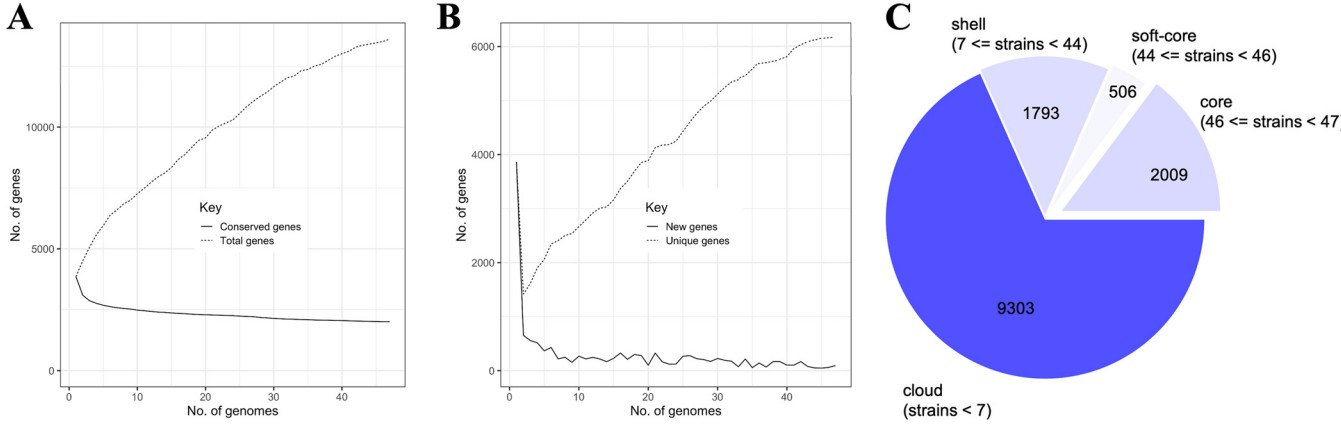

**FIG 1** (A) Conserved and total gene numbers in the pangenome. (B) Novel and unique gene numbers in the pangenome. These graphs indicate how the pangenome varies as genomes are added. (C) Pangenome pie chart showing the number of core and accessory genes. Accessory genes were divided into soft-core, shell, and cloud genes.

based resistance (21, 22). The locus containing the genes involved in the production and the assembly of the polysaccharide capsule (KL) is chromosomally carried and typically ranges from 20 to 35 kb in size. There is a high diversity among *A. baumannii* KL types, and so far, 237 have been identified (15).

In this study, we assessed the phenotypic and genetic diversity levels of 43 new *A. baumannii* modern clinical isolates, and we compared them to the established type strains of *A. baumannii* in the field.

## RESULTS

**Genome comparison of *A. baumannii* isolates.** A new strain collection containing multidrug-resistant, extensively drug-resistant, pandrug-resistant, and carbapenem-resistant modern clinical isolates of *Acinetobacter baumannii*, isolated from 2014 to 2017 from various infection sites, was recently published (23). Comparison of the whole-genome sequences (WGS) of these 43 modern clinical isolates with four established strains of *A. baumannii* identified a core genome containing 2,009 genes (Fig. 1), representing 14.76% of coding sequences (CDS) from the pangenome of 13,611 CDS. The more *A. baumannii* genomes that are analyzed, the more unique genes that are identified (Fig. 1), while the number of conserved genes decreases. These findings confirm the important variability of *A. baumannii* bacteria, pointing toward a still-open pangenome. Therefore, every time a new isolate of *A. baumannii* is sequenced, most likely (a) novel gene(s) may be identified.

The genomes of the modern strains were compared to the established strains of *A. baumannii* regarding their respective sequence type (ST; both Pasteur and Oxford schemes), capsule locus type (KL), and lipooligosaccharide locus (OCL) types. Twelve ST groups are identified in the modern collection, with ST2 as the most prevalent (25/43) followed by ST636 (6/43), ST1 (4/43), ST85 (2/43), ST78 (2/43) and ST604, ST215, ST158, and ST10 (one isolate each) (Fig. 2). Established strains belong to ST1 (AB5075-VUB), ST52 (ATCC 19606-VUB), ST437 (ATCC 17978-VUB), and ST738 (DSM30011-VUB). The most frequent capsule type was KL40 (10/43) followed by KL9 (6/43), KL3 (5/43), KL2 (4/43), KL13 and KL4 (3/43), KL22 and KL6 (2/42), and KL125, KL124, KL81, KL58, KL18, KL12, KL10, and KL7 (one isolate each) (Fig. 2). We confirm that established strains ATCC 19606-VUB and ATCC 17978-VUB both belong to KL3, while DSM30011-VUB belongs to KL47 and AB5075-VUB belongs to KL25 (24–26). In this regard, established strains ATCC 19606-VUB and ATCC 17978-VUB, both KL3, are more representative to a certain extent of modern clinical isolates than DSM30011-VUB (KL47) and AB5075-VUB (KL25), which were the sole members of their respective KLs. The typing of the locus containing the OCL of the LOS reveals a high occurrence of OCL1 (31/43) followed by OCL2 (7/43), OCL6 and OCL3 (2/43), and OCL5 (1/43). We have detected and confirmed

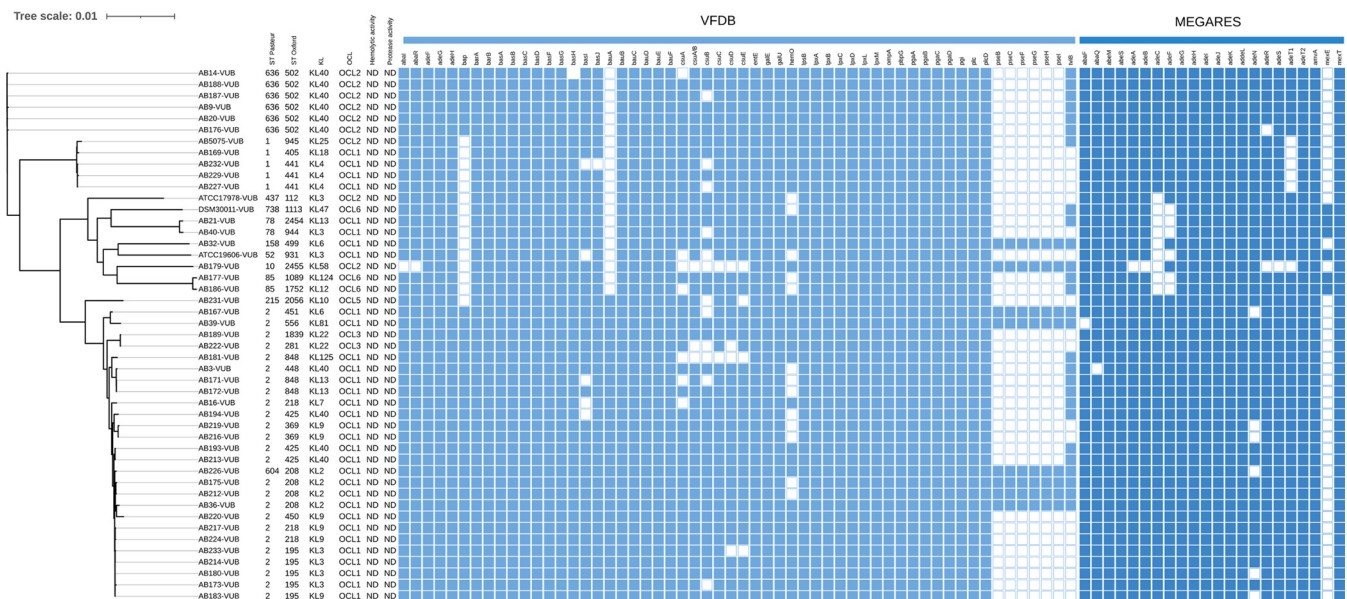

**FIG 2** A phylogenetic tree of 43 modern clinical isolates and four established strains of *A. baumannii* with depiction of ST (Pasteur scheme) and ST (Oxford scheme), KL, OCL, hemolytic activity, protease activity, and virulence genes as detected using VFDB 2022 (light blue squares) and MEGARes 2.0 (dark blue squares) databases. KL, capsule locus type; OCL, lipooligosaccharide outer core locus type; ND, not detected.

that established strains belong to OCL2 (AB5075-VUB and ATCC 17978-VUB), OCL3 (ATCC 19606-VUB), and OCL6 (DSM30011-VUB) (17, 27). So far, phylogeny along with genetic analyses shows that except for AB5075-VUB, the established strains ATCC 17978-VUB, ATCC 19606-VUB, and DSM30011-VUB cluster only with the clinical isolates of rare STs. Even though AB5075-VUB is the sole member of KL25, it clusters closer to the modern clinical isolates of *A. baumannii*.

Since we recently described the resistance of examined clinical isolates of *A. baumannii* to conventional antibiotics (23), here we focus on virulence factor-encoding genes. To explore the genetic background of the virulence factors, we employed two complementary databases: the Virulence Factor Database (VFDB) 2022 (28) and MEGARes 2.0 (29). Among other genetic differences (Fig. 2), using the VFDB 2022 (see Materials and Methods) with a standard (90%) coverage threshold applied, a variety of genes from operons encoding RND efflux pumps involved in virulence are not detected in strains ATCC 19606-VUB, ATCC 17978-VUB, and DSM30011-VUB. Using the same parameters, a key gene involved in biofilm formation (*bap*) is undetected in all four established strains (Fig. 2). However, by lowering the threshold to 86% coverage, further investigations detect the *bap* gene in ATCC 19606-VUB and ATCC 17978-VUB (see Discussion). Employing the MEGARes 2.0 database, we found that only the established strain AB5075-VUB carried all three RND efflux pumps, *adeIJK*, *adeABC*, and *adeFGH*, while DSM30011-VUB, ATCC 17978-VUB, and ATCC 19606-VUB had an incomplete *adeABC* (lack of *adeC* gene in all three strains) and/or *adeFGH* (lack of *adeF* gene in DSM30011-VUB and ATCC 19606-VUB) operon. Besides these three established strains, only six modern clinical strains of *A. baumannii* contain one or more incomplete operons encoding RND efflux pumps.

A BLAST search facilitated by ABRicate confirmed the presence of the gene encoding H-NS in all the clinical isolates and established strains of *A. baumannii* used in this study.

**Classification of macrocolony morphologies of *A. baumannii*.** A high level of genetic heterogeneity in our validated (23) collection of modern clinical isolates was observed (Fig. 1 and 2). To test whether this observation correlates with different levels of phenotypic heterogeneity, we spotted each strain on different blood- and milk-derived media and observed the potential hemolytic or secreted protease activities (30), as well as the ultrastructure of macrocolonies (Fig. 3). The clinical isolates and the

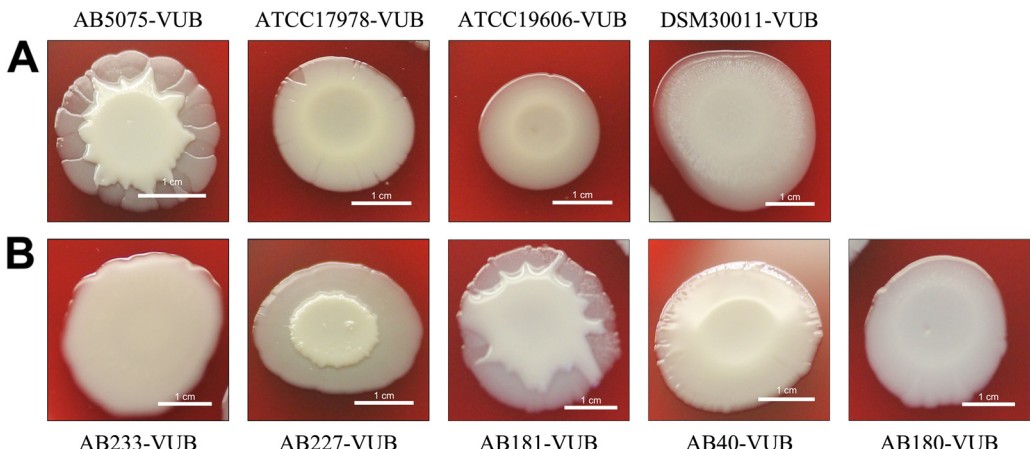

**FIG 3** Diversity of the macrocolony types. The established strains (A) compared to modern clinical isolates (B) grown on Columbia agar plates with 5% sheep blood. Bar, 1 cm.

established strains do not show any detectable hemolytic or/and protease activities under the tested conditions, which is a common trait among *A. baumannii* bacteria (31) (Fig. 2). Established strains AB5075-VUB, DSM30011-VUB, ATCC 17978-VUB, and ATCC 19606-VUB represent this common phenotypic trait, despite the high genetic diversity observed, confirming that the tested modern clinical isolates did not acquire increased hemolytic or secreted protease activities.

Analysis of the macrocolonies' morphology highlights an important diversity level concerning modern clinical isolates compared to established strains. As shown in Fig. 3 and Table 1 and in Fig. S1 in the supplemental material, we observe six different categories of macrocolony ultrastructure, with only three types (macrocolony type C [MTC] [AB5075-VUB], macrocolony type E [ATCC 17978-VUB and ATCC 19606-VUB], and macrocolony type F [DSM30011-VUB]) from established strains showing a counterpart among the modern clinical isolates. We could identify three constitutive mucoid strains, all of them being modern clinical isolates, in the macrocolony type (MT) A group of *A. baumannii* isolates (Fig. 3B). The colonies of the isolates with constitutive mucoid phenotype have a circular shape and smooth margin and a typical "oil-on-water" phenotype. The macrocolony type with an "egg on a pan" morphology (circular with a raised opaque center on a translucent regular base) appears in four strains (modern clinical isolates), representing the MTB group (Fig. 3B). The most abundant type of colony, with 19 strains including one established strain (AB5075-VUB), was MTC. MTC has a circular shape, an opaque raised center branching to the edge of the macrocolony, and an irregular translucent base (Fig. 3A). The MTD group included 14 modern clinical isolates and produced circular colonies with a raised center with most of the opaque top reaching over the edge of the macrocolony. Two established strains

**TABLE 1** Categories of macrocolonies for established strains and modern clinical isolates[a]

| Macrocolony type (MT) | Example isolate | Other isolate(s) |
|---|---|---|
| A | AB233-VUB | AB3-VUB, AB173-VUB |
| B | AB227-VUB | AB16-VUB, AB32-VUB, AB186-VUB |
| C | AB181-VUB | AB20-VUB, AB21-VUB, AB39-VUB, AB169-VUB, AB171-VUB, AB172-VUB, AB175-VUB, AB176-VUB, AB177-VUB, AB187-VUB, AB188-VUB, AB193-VUB, AB194-VUB, AB217-VUB, AB226-VUB, AB229-VUB, AB232-VUB, AB5075-VUB |
| D | AB40-VUB | AB9-VUB, AB14-VUB, AB36-VUB, AB167-VUB, AB179-VUB, AB183-VUB, AB189-VUB, AB214-VUB, AB216-VUB, AB219-VUB, AB220-VUB, AB222-VUB, AB231-VUB |
| E | AB180-VUB | AB213-VUB, AB224-VUB, ATCC 17978-VUB, ATCC 19606-VUB |
| F | DSM30011-VUB | AB212-VUB |

[a]Six categories (MTA to MTF) are based on the colony morphology phenotype. Each category is represented by one isolate, which is depicted in Fig. 3.

(Fig. 3B) and three modern clinical isolates (five strains in total) belong to the MTE group with a circular shape and undulating edge, flat center, and visible inner ring. The MTF group consists of two strains, including the established strain DSM30011-VUB. MTF has a flat macrocolony with a visible inner circle surrounded by a more translucent outer zone.

Hence, the various shapes of the macrocolonies allow division of the tested *A. baumannii* isolates into six macrocolony type groups with one representative isolate depicted in Fig. 3 (Table 1). A corresponding image for each strain analyzed in this study can be found in Fig. S1.

**Capsule production in *A. baumannii* strains.** As the mucoid phenotype can reflect the presence of an abundant polysaccharide capsule surrounding bacteria (32), we used a density gradient assay to assess the encapsulation level of all *A. baumannii* isolates (Fig. 4) (33). In this phenotypic assay, low-density bacteria have a high capsulation level, while denser bacteria have lower capsulation levels (Fig. 4A). A high degree of heterogeneity in the density of *A. baumannii* bacteria, crossing the full range from low to high density levels is observed (Fig. 4B, C, and D). Two established strains show high densities (ATCC 17978-VUB and DSM30011-VUB), while two established strains (ATCC 19606-VUB and AB5075-VUB) are characterized by a medium density level. Despite having low capsulation levels, ATCC 19606-VUB consistently shows a lower density than that of ATCC 17978-VUB. Despite two established strains (ATCC 17978-VUB and DSM30011-VUB) having a high-density phenotype and two (ATCC 19606-VUB and AB5075-VUB) having a medium-density phenotype, none of the established strains exhibited a low-density phenotype. Considering that eight (Fig. 4D) clinical isolates of *A. baumannii* were of low density (strong capsule producers), this highlights the absence of a model strain representing this phenotype among the established strains. However, the density gradients cannot be fully correlated with macrocolony types due to their high heterogeneity and certainly cannot be represented by an established strain. In addition, four clinical isolates (AB175-VUB, AB179-VUB, AB213-VUB, and AB220-VUB) are divided into two fractions within the same density gradient (Fig. 4D) (34).

In conclusion, 8 out of 43 modern clinical isolates are not represented by the here-tested established strains of *A. baumannii* in terms of the density gradient phenotypes.

To directly observe capsule deposition and thickness at high resolution and at the single-cell level, we combined capsule labeling with transmission electron microscopy (TEM) (Fig. 5). As expected, low-density bacteria are surrounded by a thicker capsule than that around high-density bacteria (Fig. 5B). Concerning the established strains, we confirm AB5075-VUB to be the more capsulated established strain tested in this study. We could not detect any obvious capsule deposition on the ATCC 17978-VUB strain, and the ATCC 19606-VUB strain shows a weak and heterogeneously deposited layer under the tested conditions (Fig. 5A). This is in line with the established strain ATCC 19606-VUB being less dense (medium density) than the established strain ATCC 17978-VUB (high density) in our density gradient assay (Fig. 4B and D). Among 13 high-density (less capsulated) modern clinical isolates (Fig. 4D), two isolates, AB180-VUB and AB183-VUB, show less abundant capsule deposition in the TEM micrograph as well (Fig. 5B). Of the eight least dense clinical isolates, the isolates AB3-VUB, AB193-VUB, AB213-VUB, and AB233-VUB were confirmed to produce a high abundance of capsule deposits (Fig. 5B). Accordingly, strains AB36-VUB and AB39-VUB show a medium density level and have an intermediate capsule deposition. Taken together, these data show that bacterial density correlates with capsule abundance of *A. baumannii* bacteria as shown before (33) and that the established strains do not show all the heterogeneity observed in the modern clinical isolates tested regarding capsule production level.

**Constitutive mucoid strains are not hypervirulent.** As constitutive mucoid strains are observed only among the group of modern clinical isolates, we wondered if this phenotype could be associated with higher *in vivo* virulence. *Galleria mellonella* larvae were infected with highly, medium-, and weakly capsulated isolates, while the AB5075-VUB established strain was used as a positive control of virulence (35). Concerning the constitutive mucoid isolates, while AB213-VUB is highly virulent, the constitutive

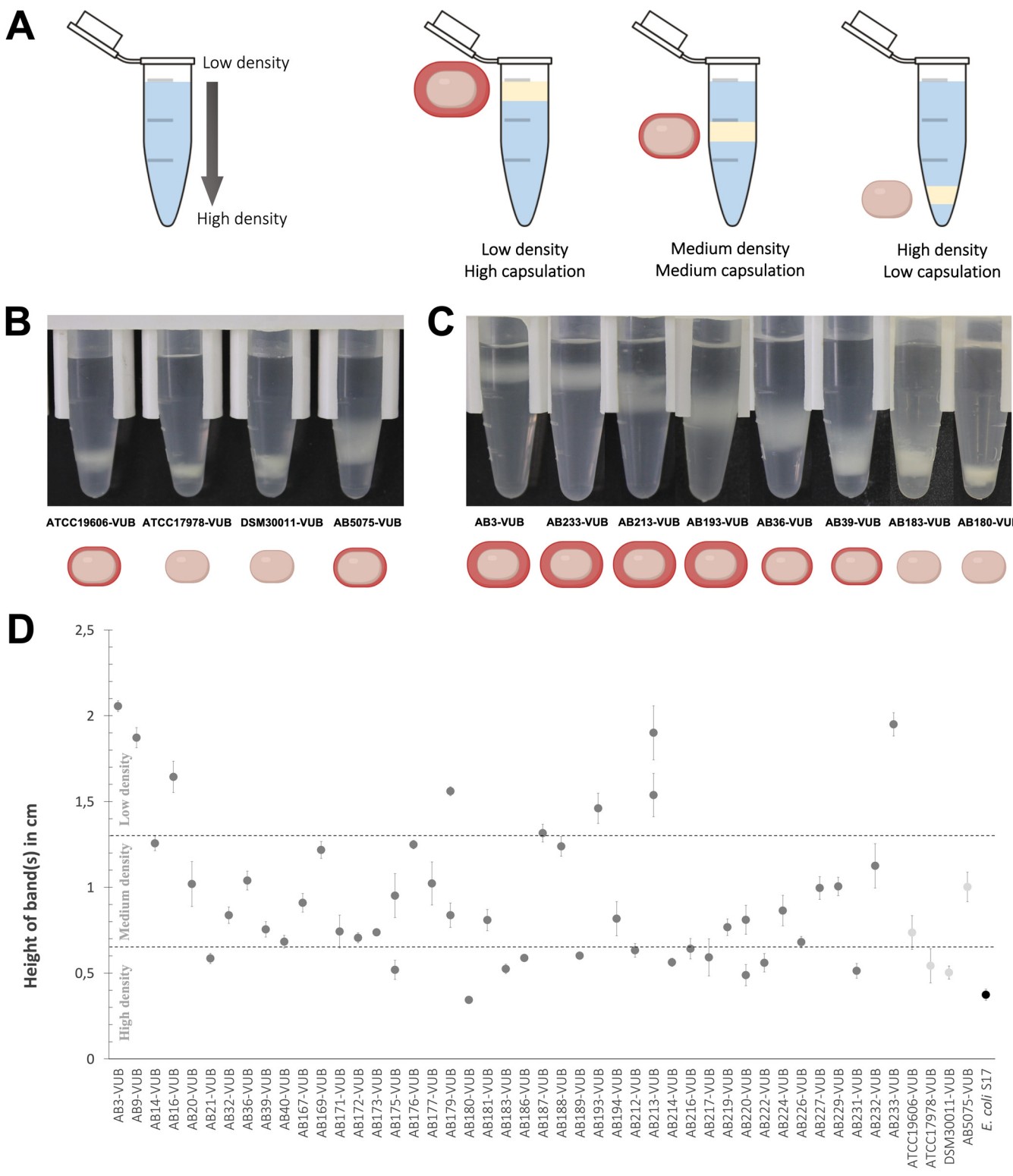

**FIG 4** Quantification of the density of bacterial cells measured in a gradient colloidal silica. (A) Depiction of the density gradient measuring principle. (B and C) Capsulation level of the established strains (B) and comparison of various levels of capsulation of the modern clinical strains (C) of *A. baumannii*. (D) Height of bands corresponding to the cellular density of each isolate divided (from top to bottom) into low, medium, and high density. The list and the capsulation level are summarized in Table S2 in the supplemental material. The standard deviation was calculated from the biological triplicates.

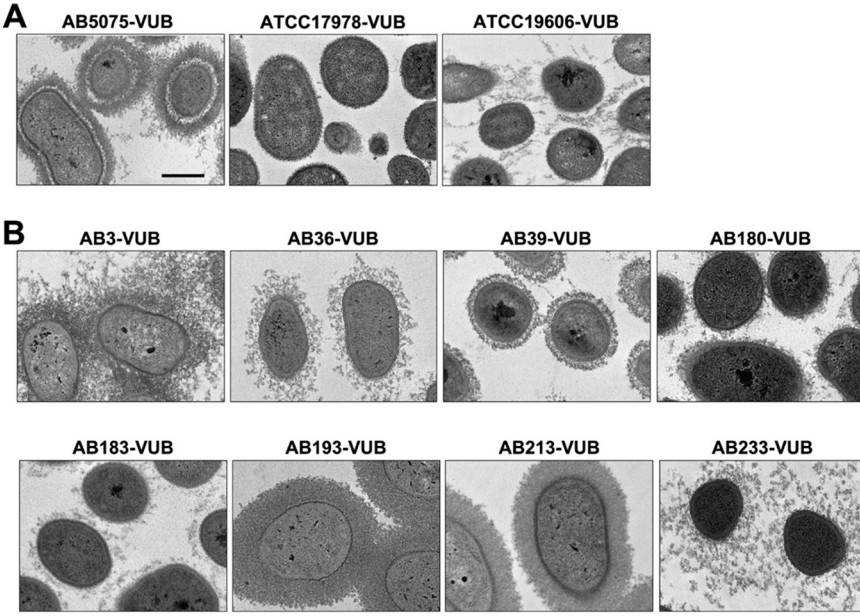

**FIG 5** Direct capsule visualization using TEM. (A) Established strains. (B) Eight modern clinical isolates of various cell densities. Bar, 500 nm.

mucoid isolates AB3-VUB and AB233-VUB are weakly virulent (Fig. 6). The medium-capsulated isolate AB39-VUB is highly virulent and killed 100% of the larvae within 1 day, being even more virulent than AB5075-VUB, while the medium-capsulated AB36-VUB showed an intermediate virulence level. Interestingly, the low-capsulated isolates (AB183-VUB and AB180-VUB) are less virulent, AB180-VUB being avirulent under the tested conditions (Fig. 6). These data show that a constitutive high capsulation and mucoid phenotype does not correlate with a higher virulence potential in *A. baumannii*. However, this highlights that a low capsulation level characterizes weakly virulent phenotypes, confirming that *A. baumannii* bacteria require capsule deposition for full virulence. This is in agreement with previous studies showing that genetically manipulated capsule-deficient *A. baumannii* strains have decreased virulence levels (33, 36). The virulence does not directly correlate with a specific KL or OCL type (Fig. 2 and 6); hence, we can conclude that the KL or OCL is not solely responsible for *A. baumannii*

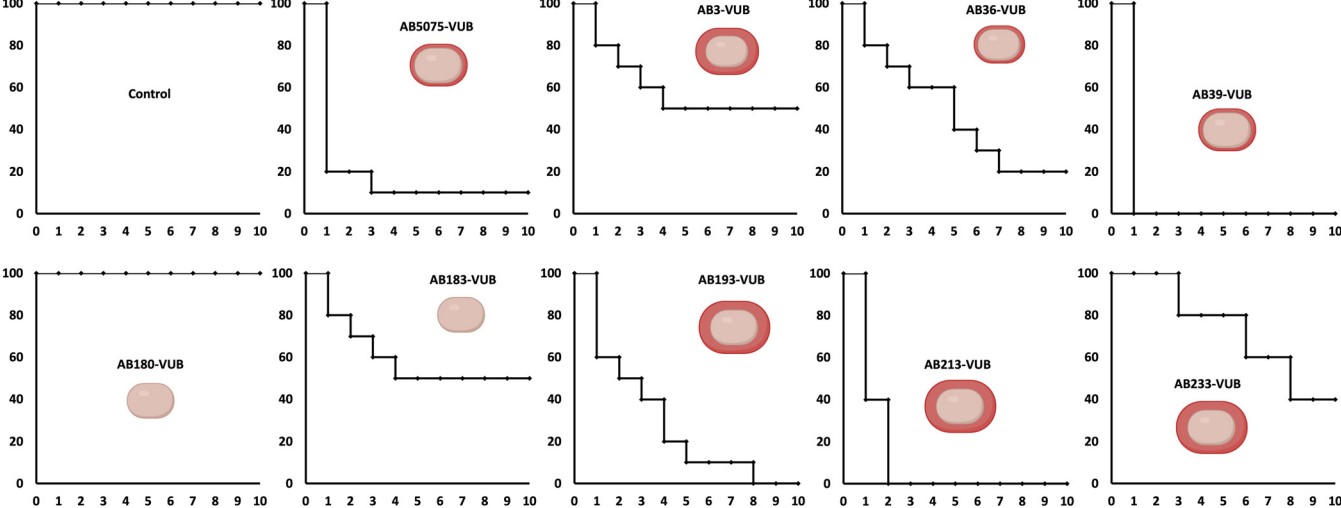

**FIG 6** *In vivo* virulence of modern *A. baumannii* isolates determined in *G. mellonella* model. *x* axis, days after inoculation; *y* axis, percentage of living *G. mellonella*. The control condition is PS without bacteria.

virulence under the tested conditions, which might require multifactorial influence. Taken together, these observations show that the established strain AB5075-VUB is a good bacterial model to study the virulence potential in *G. mellonella*; however, we detected a strain with even higher virulence under the tested conditions (hypervirulent AB39-VUB).

**Natural transformation of *A. baumannii* isolates.** We next assessed the ability of natural transformation of all the *A. baumannii* strains. AB5075-VUB was used as a positive control due to its verified competence (37). We have categorized the strains regarding the level of competence (Table S1). These results show variability in the natural transformation potential of the clinical isolates and established strains of *A. baumannii*, as previously observed (11, 38).

## DISCUSSION

The genotypes and phenotypes of our collection of 43 modern clinical isolates and 4 established strains of *A. baumannii* were assessed, compared, and evaluated. We identified 12 different ST groups in this new collection of 43 modern clinical isolates. While the established strain AB5075-VUB belongs to less frequently occurring ST1 (Pasteur), the broadly used strains ATCC 19606-VUB (ST52), ATCC 17978-VUB (ST437), and DSM30011-VUB (ST738) do not reflect the current trends of ST in clinical settings, at least when referred to our collection here tested, but instead rare sequence types. The lack of representation of the ST of the clinical isolates is expected only in the case of DSM30011-VUB, as this strain has an environmental origin (39). The ST2 group, previously described as one important and clinically relevant group, is also the most widely disseminated ST among the complete and draft genomes available (40). Most of the modern clinical isolates of our collection belonged to ST2, yet none of the established strains represented this sequence type. In the future, new widely used strains of *A. baumannii* should be considered.

Our phylogenetic analysis (Fig. 2) identified several clades, mostly in accordance with the ST of the isolates. The exceptions are the clade consisting of isolates which belong to ST2 with AB226 (ST215) and AB231 (ST604) and the clade with established strains (DSM30011-VUB, ATCC 17978-VUB, and ATCC 19606-VUB) which includes 9 isolates of 7 different STs. The clade with established strains DSM30011-VUB, ATCC 17978-VUB, and ATCC 19606-VUB consists of clinical isolates with undetectable *adeC* and *bauA* genes (see Materials and Methods); therefore, these established strains can still represent a minor proportion of the current clinical isolates of *A. baumannii* to a certain extent. On the other hand, of four established strains, AB5075-VUB is the most similar to the modern clinical strains of *A. baumannii* in regard to OCL and virulence genes. When considering AB5075-VUB capsule production, its phenotype resembles phenotypes of clinical isolates with lower production. Nevertheless, similarly to other established strains, *adeC* and *bauA* genes are not detected in AB5075. Lack of the combination of these two virulence genes is also detected in 10 modern clinical isolates, proving that AB5075-VUB still shares some characteristics with a subset of circulating modern clinical *A. baumannii* strains. The listed virulence genes in our study account for only a small amount of the virulence genes as only those genes known to date can be detected; likely, novel virulence genes remain to be identified. Notably, regardless of the observed discrepancy in the detection of the *adeFGH* operon using VFDB 2022 and MEGARes 2.0 databases, the absence of *adeFGH* was described previously (41, 42).

Capsule heterogeneity is a hallmark of *A. baumannii* bacteria. The surface polysaccharides play key roles in the fitness and virulence of *A. baumannii* and protect it from the environment, increase resistance to antimicrobial compounds, and help to evade the host immune system (43, 44). There are more than 237 KL types identified so far (15), with additional ones expected to be found because of the genetic heterogeneity of these bacteria. As the capsular polysaccharide (CPS) is a potential target for therapeutical agents and vaccines (45), this diversity should be surveyed in modern and

circulating clinical isolates. Among the 43 modern clinical isolates of *A. baumannii* here characterized, KL40 (10/43) dominated while KL3 was detected in five isolates.

In our study, the most prevalent type of locus encoding outer core lipooligosacchar-ide (OCL) is OCL1 (31/43). However, none of the established strains represents this most prevalent OCL type as they belong to OCL2 (AB5075-VUB and ATCC 17978-VUB), OCL3 (ATCC 19606-VUB), and OCL6 (DSM30011-VUB), suggesting their rarefaction in following the current epidemiological trends. Despite the proven relevance of the clini-cal isolates of our collection (collected as problematic and modern nosocomial iso-lates), we cannot rule out any bias to a certain extent and other clinically relevant *A. baumannii* isolates should be considered (see "Conclusion" below).

The *bap* gene encodes a protein required for the formation of three-dimensional biofilm towers and water channels on abiotic and biotic surfaces such as polypropyl-ene, polystyrene, and titanium (46). Bap protein is also involved in the adherence of *A. baumannii* to human bronchial epithelial cells and human neonatal keratinocytes (46). Interestingly, the *bap* gene can be disrupted due to mutations (47). The *bap* gene is not detected in two established strains (DSM30011-VUB and AB5075-VUB), which could influence the studies exploring biofilm properties using these established strains. However, *bap* was found in ATCC 17978-VUB and ATCC 19606-VUB with 86% gene cov-erage and 97% nucleotide identity (VFDB 2022), as well as in three clinical isolates (AB40-VUB, AB32-VUB, and AB21-VUB); therefore, it was assigned as undetected in Fig. 2 as the identity falls under the applied threshold of 90%. The lower identity of the *bap* gene in the strains ATCC 17978-VUB and ATCC 19606 and the three clinical isolates (AB40-VUB, AB32-VUB, and AB21-VUB) shows that different thresholds result in positive detection of *bap*. However, the difficulties with detecting the *bap* gene in some strains and isolates might be also attributed to a limited set of reference genes in the data-bases, while the *bap* gene is variable (9).

Moreover, 20 out of 47 examined strains including four established strains do not carry detectable *bauA*, encoding iron-regulated outer membrane protein BauA (see Materials and Methods). BauA protein provides protection against sepsis caused by *A. baumannii* and was also identified as a vaccine candidate (48, 49). Interestingly, six strains (including two established strains, DSM30011-VUB and ATCC 19606-VUB, and four clinical isolates, AB177-VUB, AB186-VUB, AB40-VUB, and AB21-VUB) encode mem-brane fusion protein MexE protein, part of the RND efflux pump from *Pseudomonas aeruginosa* (50). Notably, the very same six strains were lacking a gene encoding mem-brane fusion protein AdeF from the AdeFGH RND efflux pump, possibly restoring its function as AdeF and MexE share 50% amino acid identity. However, using the VFDB 2022 database, all clinical isolates and established strains contain a detectable *adeFGH* operon. Further investigations regarding the consequences of detected mutations and the functionality of these proteins are required.

Deficiency of the clinical isolates and established strains in hemolytic and protease activities is in agreement with the fact that only some species of the *Acinetobacter* ge-nus, e.g., *Acinetobacter haemolyticus* (51), are capable of hemolysis. Even despite this shared absence of a phenotypical trait, there is a high level of diversity of macrocolo-nies, where established strains represent only three variants out of six types observed in the clinical strains, while strain DSM30011-VUB is one of two members of the MTF group. Notably, the strain DSM30011-VUB is also the sole representative of *A. baumannii* of environmental origin in this study (39); therefore, it will be interesting to test other environmental isolates in this context. Three modern clinical isolates show a constitutive mucoid phenotype that is not observed in the established strains. Isolates belonging to the same phylogenetic clusters (Table 1 and Fig. 2) do not have the same ultrastructure of the macrocolony, showing a high diversity even within the same phylogenetic group. To be noted, also external events such as gene disruption can affect the phenotype of the macrocolony, as was observed in AB5075-UW by Pérez-Varela et al. (52) by disrupting a *relA* ortholog (*ABUW_3302*) using transposon insertion. However, phase variation, a

hallmark of *A. baumannii* bacteria, might be involved in defining these macrocolony types (34), with different levels of phase variation correlating with specific MT groups.

We observe a high diversity in the production of the capsular polysaccharide using direct and indirect visualization methods. We show a correlation between capsule abundance and density levels, but correlation does not mean causation. We cannot rule out the possibility that the capsule type *per se* (and not only the capsule abundance) or/and other factors influence the density of *A. baumannii* bacteria, even if we cannot detect a correlation between cellular density and capsule type in our study. Interestingly, four clinical isolates (AB175-VUB, AB179-VUB, AB213-VUB, and AB220-VUB) are divided into two fractions within the same test tube in the density gradients (Fig. 4D). This observation suggests diversity in the production of CPS even within the same isolate, possibly pointing toward a high frequency of phase variation leading to phenotypic heterogeneity. There are multiple factors leading to phase variation such as accumulation of an extracellular signal (34), copy number of a plasmid-borne resistance-encoding integron (53), or stress-related disruption of a gene within the capsule synthesis pathway (33).

The fact that widely used strains show low to medium encapsulation degrees adds an additional reason why established strains should be carefully used and argues in favor of considering the strain AB5075-VUB as the most representative, to some extent, bacterial model out of the four established strains tested in this study.

The *Galleria mellonella* infection model proved itself as a valuable source of information on the virulence level of *A. baumannii*, showing the AB5075-VUB strain to be more virulent than ATCC 19606-VUB and ATCC 17978-VUB under the tested conditions (54). The established strains ATCC 19606 and ATCC 17978 as well as the environmental strain DSM30011 are also virulent in the *G. mellonella* model of infection (55–57). These results support the use of modern clinical isolates for the study of virulence; hence, other highly virulent isolates such as the hypervirulent *A. baumannii* LAC-4 strain (58) should be considered as well. Assessing the virulence level of the LAC-4 isolate versus the hypervirulent strain AB39-VUB here described can promote AB39-VUB as an alternative model of hypervirulence. The virulence in the *G. mellonella* model varied for tested clinical isolates of our collection, and while a constitutive mucoid phenotype does not correlate with higher virulence, we confirm using modern clinical isolates that low capsule production impedes full virulence in *A. baumannii*. These observations do not confirm the results of Shan et al. (59), who concluded that the mucoid *A. baumannii* isolates were more hypervirulent than the nonmucoid strains. This discrepancy may point toward a multifactorial background of hypervirulence. The signal(s) or conditions regulating capsule (hyper)production remain to be determined. Notably, the isolates AB193-VUB and AB213-VUB with higher integrity of the CPS layer (Fig. 5) show higher virulence in *G. mellonella* (Fig. 6) than do isolates with dispersed CPS.

Competence for natural transformation is one of the ways of horizontal gene transfer that *A. baumannii* uses to acquire extracellular DNA from the environment and incorporate it into its own genome via homologous recombination (60). We observed that natural transformation potential varies within clinical isolates of *A. baumannii*. This diverse trend is copied by established strains AB5075-VUB and DSM30011-VUB, which are successfully transformed while ATCC 17978-VUB is not and ATCC 19606-VUB shows intrinsic resistance to apramycin. There are multiple factors influencing the ability of *A. baumannii* to be naturally transformed, such as the presence of H-NS (37), which is present in each case within the isolates tested in our study. A recent study by Godeux et al. (61) reported transformation followed by spontaneously occurring recombination events in a mixed bacterial population where the acquisition of large resistance islands such as AbaR1 and AbaR4 facilitated carbapenem resistance. However, the major role is played by type IV pilus genes (62), which are growth phase dependent (63) and might be the explanation of some isolates and strains not being transformed. This is pointing out further physiological diversity within *A. baumannii* clinical isolates and established strains.

Taken together, this shows the insufficiency of established strains in systematically capturing the phenotypical and genotypical diversity of modern clinical isolates. The study of genetic features that are not shared by the majority of *A. baumannii* isolates (soft-core, shell, and cloud genes) (64) may lead to novel discoveries. However, the global impact of the clinical importance (drug therapies or target-driven drug discoveries) will suffer from linking the specific genes and features only to specific strain(s). Taking peculiarities of *A. baumannii* bacteria into account, a strategy to rationally select a relevant strain(s) as the bacterial model in a specific study associated with a targeted biological question is required. In this context, the new "Acinetobase" (65) is a comprehensive database providing the community with the genotype, phenotype, and strains of *Acinetobacter* spp.

**Conclusion.** *A. baumannii* organisms are heterogenous bacteria, at both the genetic and phenotypic levels. In this study, we characterized 43 modern clinical isolates from different phylogenetic groups and four established strains with common but also very different features. Therefore, the historically established strains (AB5075, ATCC 17978, ATCC 19606, and DSM30011) tested in our study do not cover the whole heterogeneity found in the modern clinical isolates of *A. baumannii*. The studies previously published using these established strains built a strong state of the art in the *A. baumannii* field and beyond, showing their usefulness. However, the data presented in our study show that the specific use of one or only a limited subset of established strains can hinder important processes characterizing clinically relevant isolates. As an answer to that identified pitfall, we propose a variable collection of modern clinical isolates that are characterized at the genetic and phenotypic levels, covering the full range of the phenotypic spectrum, with six different macrocolony type groups, from avirulent to hypervirulent phenotype, and with noncapsulated to hypermucoid strains, with intermediate phenotypes as well. This will allow selection of an appropriate strain rationally, facilitated by the new Acinetobase (65) platform, which suits the needs of ongoing studies with a particular biological question and will help to overcome bias caused by the use of single nonmodern strains and will provide genotypic and phenotypic characterizations and the strains of *Acinetobacter* spp. themselves. This is especially important for new antimicrobial screening purposes, for which conserved targets among a significant proportion of problematic *A. baumannii* isolates are a prerequisite. While strain-specific observation remains interesting *per se*, in the context of such a drastic heterogeneity, any newly identified targets, antimicrobial compounds, or fundamental observations deserve to be tested on diverse relevant *A. baumannii* isolates. Rather, a rational choice of characterized isolates, related to the biological question investigated, will help to better understand these peculiar bacteria. Nevertheless, we do also acknowledge limitations concerning our strain collection such as geographical bias and the number of isolates examined as well as the restriction of our research to specific genetic and phenotypic approaches. Further studies involving additional bacterial isolates from various geographical sites and hosts, together with microscopic imaging, proteomic, and expression analyses, will be required.

## MATERIALS AND METHODS

**Modern clinical isolates, widely used strains, and phylogeny inference.** A nonredundant collection of 43 modern (isolated from 2014 to 2017) clinical isolates from the National Reference Center for Antibiotic-Resistant Gram-Negative Bacilli (CHU UCL-Namur) and four established strains, ATCC 17978-VUB, ATCC 19606-VUB, DSM30011-VUB, and AB5075-VUB, were studied. The established strains AB5075, ATCC 17978, and ATCC 19606 are non-MDR (except AB5075) strains of clinical (osteomyelitis, meningitis, and urine, respectively) origin (14, 66), while DSM30011 is a non-MDR, environmental strain obtained from plant microbiota (39). The established strains were designated with the extension "-VUB" to distinguish the strains and their sequences examined within our study. The strain ATCC 17978 used in this study was PCR screened for the presence of the 44-kb locus AbaAL44 (67) in 14 randomly picked single colonies, confirming its AbaAL44$^+$ genotype. The selected isolates were whole genome sequenced using short reads (Illumina) combined with long reads (Oxford Nanopore Technologies [ONT]) in our previous study (23).

**Natural transformation of the isolates.** The natural transformation of the clinical isolates and the established strains of *A. baumannii* was assessed by starting an overnight bacterial culture from −80°C stock in 5 mL of LB (37°C at 160 rpm). The culture was diluted 1:100 in a 2-mL microtube (10 μL of the

bacterial culture plus 990 $\mu$L of tryptone solution at 5 g/L). Then, 3 $\mu$L of the diluted culture was mixed with 3 $\mu$L of plasmid (apramycin resistance and *sfGFP* genes [33]) DNA (100 ng/$\mu$L) in a 1-mL microtube. Subsequently, 3 $\mu$L of the mix of culture and plasmid DNA was transferred to a 2-mL microtube containing 1 mL of tryptone at 5 g/L and 2% agar. A 3-$\mu$L amount of diluted bacterial culture without plasmid DNA was used as a negative control.

After 6 h of incubation at 37°C, 100 $\mu$L of tryptone solution at 5 g/L was added to the microtube with the culture and plasmid DNA and vortexed gently. A 50-$\mu$L amount of the suspension was plated on LB agar plates containing apramycin (50 $\mu$g/mL) to select transformants. The negative control was plated on apramycin-LB agar plates, too. Colonies were counted after overnight incubation at 37°C.

**Hemolytic and protease activities.** To assess the potential hemolytic activities of the different *A. baumannii* isolates, we spotted 5 $\mu$L of an overnight (O/N) culture of bacteria previously grown in LB medium for 16 h at 37°C under constant agitation (175 rpm) on 4 different blood agar plates: (i) Columbia agar with 5% horse blood, (ii) Columbia agar with 5% sheep blood, (iii) Trypticase soy agar II with 5% horse blood, and (iv) Trypticase soy agar II with 5% sheep blood, all purchased from BD (Becton, Dickinson and Company, Franklin Lakes, NJ). To test for secreted protease activity, we used the same approaches as described above and spotted the 5 $\mu$L of bacteria on LB agar plates containing 2% skim milk powder for microbiology (Sigma-Aldrich/Merck KGaA, Darmstadt, Germany). Plates were incubated at 25°C and monitored for hemolytic and protease activities after 1, 2, and 6 days of incubation.

**Phylogenetic analysis.** The maximum-likelihood tree depicting the relatedness of the isolates was constructed from assembled complete genomes using predicted open reading frames obtained by Prokka (https://github.com/tseemann/prokka) as an input for the core-genome alignment created using Roary (https://github.com/sanger-pathogens/Roary). RAxML (https://github.com/stamatak/standard-RAxML) was used for the calculation of the phylogenetic tree using the general time reversible with optimization of substitution rates under gamma model of rate heterogeneity method supported by 500 bootstraps. The phylogenetic tree was visualized in iTOL (https://itol.embl.de/).

**Genotypic characterization.** The resistance and virulence genes were detected using ABRicate (https://github.com/tseemann/abricate) employing VFDB 2022 (28) and MEGARes 2.0 (29) databases, respectively, with a 90% threshold for both gene identity and coverage. The typing of capsule-encoding loci (KL) and lipooligosaccharide outer core loci (OCL) was determined using Kaptive (68, 69) after manual curation of the corresponding loci by mapping the short reads on the anticipated reference sequence of the KL using Geneious R9 (Biomatters, New Zealand). The presence of the gene encoding H-NS was detected using ABRicate (https://github.com/tseemann/abricate) with a custom database containing H-NS of AB5075-UW (accession number CP008706.1) as a reference sequence.

**Macrocolony morphology.** Five microliters of overnight bacterial suspension ($\sim$1 $\times$ 10$^8$ cells) was plated on Columbia agar with 5% sheep blood purchased from BD (Becton, Dickinson and Company, Franklin Lakes, NJ). The plates were incubated noninverted for 6 days at 25°C and subsequently photographed by a Canon camera. The experiments were carried out in biological triplicates confirming the reproducibility of the macrocolony morphology assessment. The representatives of each group were selected (Fig. 3), and the corresponding macrocolonies for each strain can be found in Fig. S1 in the supplemental material.

**Capsule production.** One milliliter of overnight culture in a 1.5-mL microtube was centrifuged for 2 min at 7,000 relative centrifugal force (rcf). The supernatant was removed, and the pellet was resuspended in 1 mL of phosphate-buffered saline (PBS). Subsequently, 875 $\mu$L of PBS-resuspended bacteria was mixed with 125 $\mu$L of Ludox LS colloidal silica (30% [wt/wt] suspension in H$_2$O [Merck]) (32, 70). This mix was then centrifuged for 30 min at 12,000 rcf and immediately photographically recorded. The distance of the center of the band from the bottom of the microtube was measured. The measurements were taken in biological triplicates and statistically evaluated by calculating the standard deviation from the mean values for biological replicates. *Escherichia coli* S17 (ATCC 47055) was used as a low-capsulated control.

**TEM.** Transmission electron microscopy (TEM) was used for direct capsule visualization by labeling the capsule of 11 *A. baumannii* isolates, 8 modern clinical isolates and 3 established strains, which range from high to low densities. The fixation and staining of the bacteria were performed as described before (71). The cupule with the fixed pellet of bacteria (polymerized for 5 h) was embedded in resin and polymerized (12 h at 37°C, 48 h at 45°C, and 3 days at 60°C). The $\sim$60-nm slides of the resin were marked with acetate uranyl and placed on the electron microscopy grid.

**Virulence in *Galleria mellonella* model of infection.** TruLarv research-grade larvae of *G. mellonella* (BioSystems Technology) were stored at 15°C no longer than 5 days after arrival and were incubated for 30 min at 4°C prior to injection. Bacteria from an overnight culture were washed with physiological saline (PS) (0.9% NaCl) and diluted to approximately 1 $\times$ 10$^7$ CFU/mL. The larvae were injected with 10 $\mu$L of PS containing 1 $\times$ 10$^5$ CFU/mL of *A. baumannii* in the last left proleg using a 0.3-mL insulin syringe (BD MicroFine). Each of the nine selected strains of *A. baumannii* was injected into 10 larvae, and 10 larvae were injected with PS as a negative control. The experiments were carried out in duplicates, and the survival (assessed by keratinization and mobility) rate was evaluated each day for a period of 10 days as described before (33).

## SUPPLEMENTAL MATERIAL

Supplemental material is available online only.

**SUPPLEMENTAL FILE 1**, PDF file, 1.8 MB.

## ACKNOWLEDGMENTS

We are grateful to the URBM and URBE research groups, as well as the Electron Microscopy Service from UNamur, for the access to their equipment and their expertise. We thank Ivan Mateus from the Department of Ecology and Evolution, University of Lausanne, Lausanne, Switzerland, for the initial bioinformatical analyses. We thank Johanna Kenyon for fruitful discussions and advice regarding KL and OCL analyses. We thank the National Reference Laboratory for Monitoring of Antimicrobial Resistance in Gram-negative Bacteria, CHU Mont-Godinne, Université Catholique de Louvain (UCL), Yvoir (Belgium), for providing the modern clinical isolates and related information.

There are no competing interests to declare.

C.P., C.V.d.H., and E.R. performed phenotypical experiments. A.V., C.V.d.H., and K.N. performed bioinformatical analyses. C.W. extracted DNA for the long-read sequencing. M.B., C.P., C.V.d.H., and T.C. performed the infections of *Galleria mellonella*. T.D.P., W.D.C., and M.S. set up the long-read sequencing strategy, sample, library preparation, and sequencing. A.V. and C.V.d.H. wrote the manuscript.

This project was supported by the Flanders Institute for Biotechnology (VIB) and has received funding from the European Union's Horizon 2020 research and innovation program under the Marie Sklodowska-Curie grant agreement no. 748032. A.V. is a recipient of a junior postdoctoral fellowship of the Research Foundation – Flanders (FWO; file number 1287223N). K.N. was supported by the Erasmus + program KA131-HED.

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
