## [Reviewer comments · Microbiology Spectrum]

Microbiology Spectrum

Phenotypic characterization and heterogeneity amongst modern clinical isolates of *Acinetobacter baumannii*

Adam Valcek, Chantal Philippe, Clemence Whiteway, Etienne Robino, Kristina Nesporova, Mona Bové, Coenye Tom, Tim De Pooter, Wouter De Coster, Mojca Strazisar, and Charles Van der Henst

Corresponding Author(s): Charles Van der Henst, Vrije Universiteit Brussel

Review Timeline:

Submission Date:	August 9, 2022
Editorial Decision:	September 21, 2022
Revision Received:	November 19, 2022
Accepted:	November 22, 2022

Editor: Ayush Kumar

Reviewer(s): Disclosure of reviewer identity is with reference to reviewer comments included in decision letter(s). The following individuals involved in review of your submission have agreed to reveal their identity: Maria Dolores Alcántar-Curiel (Reviewer #2)

Transaction Report:

DOI: <https://doi.org/10.1128/spectrum.03061-22>

September 21, 2022

Prof. Charles Van der Henst
Vrije Universiteit Brussel (VUB) / Flanders Institute for Biotechnology (VIB)
VIB-VUB Center for Structural Biology
Pleinlaan, 2
Brussel 1050
Belgium

Re: Spectrum03061-22 (Established strains of *Acinetobacter baumannii* ATCC19606, ATCC17978 and AB5075 do not reflect the whole diversity of modern clinical isolates)

Dear Charles,

Thank you for submitting your manuscript to Microbiology Spectrum. Your manuscript has now been reviewed by two experts in the field and both suggest modifications to your manuscript. When submitting the revised version of your paper, please provide (1) point-by-point responses to the issues raised by the reviewers as file type "Response to Reviewers," not in your cover letter, and (2) a PDF file that indicates the changes from the original submission (by highlighting or underlining the changes) as file type "Marked Up Manuscript - For Review Only". Please use this link to submit your revised manuscript - we strongly recommend that you submit your paper within the next 60 days or reach out to me. Detailed instructions on submitting your revised paper are below.

Link Not Available

Sincerely,

Ayush Kumar

Journals Department
Reviewer comments:

Reviewer #1 (Comments for the Author):

Review mSpectrum

General comments:

In the manuscript entitled "Established strains of *Acinetobacter baumannii* ATCC19606, ATCC17978 and AB5075 do not reflect the whole diversity of modern clinical isolates", Valeck and co-authors claim to raise awareness on the use of a handful of

strains of *A. baumannii* that are shared between labs or accessible through bioresource centers.

The authors should seek help for the English and proofreading of the entire article is necessary (inappropriate use of terms (ie. proof-of-concept), some words are missing (ex. lane 136/137), the definite article "the" is often missing etc...).

The authors should also select relevant references and add several references (see specific comments).

The message conveyed by the title would have been relevant ten years ago, however, with the advent of massive genome sequencing of numerous clinical strains this statement is already widely accepted by *A. baumannii* researchers. The author should reconsider the main message, and thus the title, of the article. I recommend that the article be focused on the specific diversity of the Belgium strains collection regarding capsule and virulence traits as the authors already published more thorough results regarding the antibiotic resistance in AAC. So far, this manuscript is highly descriptive, and the author should exploit the genomic data to explain the phenotypes observed (link virulence phenotype to genotype, capsule production to genotype etc...).

Overall, the results are confirmatory of previous results published by other groups. It confirms that a phenotype variability is observed between clinical isolates for capsule production, in vivo virulence, and natural transformation. Based on the results of this manuscript, the reference strain AB5075 appears as a good candidate for experimental studies (capsule production, colony morphology, virulence in *Galleria*, genetic amenability by natural transformation).

More specific comments to help the authors improve the quality of the manuscript.

- lane 76: modify or add more suitable reference. The reference 8 is indeed too specific to a mean of decontamination (consider a review if necessary)
- lane 78: The reference 8 does not demonstrate the variability in transformation between isolates. Change to more relevant references
- lane 94: "Chromosomally encoded LOS is a major virulence component in Gram-negative bacteria". Reference is missing at this end of this statement.
- lanes 105 to 109: this information is useless for the manuscript.
- lane 78: replace in the manuscript competence by transformation (see lane 146). As written, it sounds as if natural transformation is the only horizontal gene transfer mechanism in *A. baumannii*. Be more specific.
- In the introduction, biofilm formation in *A. baumannii* is not introduced although presented in the results section. Add a paragraph to introduce this topic.
- lane 135: why is there a section "Short- and long-read sequencing" in the materials and methods as the results are already published?. The authors should remove this section (simply cite in the main text the corresponding reference).
- lane 142: again, why a section "Antimicrobial resistance genotype and phenotype" in the materials and methods as the results are already published?. The authors should remove this section.
- lane 146: the authors do not test the natural competence but natural transformation (modify in the entire manuscript competence by transformation)
- lane 161 and results section: results of the protease activity are not presented
- lane 226: replace "genotypical" by "genotypic" or "genome" comparison
- lane 266: the result on the difficulty on detecting the *bap* gene is expected as it is highly variable in size (De Gregorio et al., 2015, 10.1186/s12864-015-2136-6). Moreover, a reference is missing at the end of lane 266.
- lane 281: this sentence is clunky, consider rewriting
- lane 286: the protease or hemolytic activity results are not presented, please add some examples in supplementary materials
- lane 288: Figure 1 is mentioned erroneously
- lane 292: how reproducible is this qualitative observation? please explain in the corresponding materials and methods section
- lane 330: how is the measure performed for AB5075 given the thickness of the density band? please explain in the materials and methods
- lane 337: please specify between brackets the corresponding isolates and corresponding result on panel C (AB213-VUB).
- p16, Figure 4D: why the strain *E. coli* S17 is included in the figure but not discussed in the result section? clarify this point
- lane 365: specify that the result obtained in this study confirm previous work (Whiteway et al., 2022)
- 383: The strain AB183-VUB shares the same capsule phenotype with DSM30011 that is highly virulent for *Galleria*, please comment this seeming discrepancy (Repizo et al., 2015, 10.1371/journal.pone.0138265). The DSM30011 strain should be added to the *Galleria* assays
- lane 533: typo in ATCC19606

Reviewer #2 (Comments for the Author):

Introduction

1. Missing references 17 and 18.

Material and methods

1. Specify which type of MLST scheme has been used.
2. In the Capsule production study indicate that the assays were performed in triplicate.
3. Reference is missing in the Virulence assay in the *Galleria mellonella* infection model.
4. There is not statistical test section. The authors do not describe the statistical analysis used to quantify bacterial cell density. Please describe it.

Results

1. Line 229. Figure 2B should be Figure 1.
2. Line 943. In the caption of figure 2 the color code and the acronyms STPas, STOx, KL and OCL are not described. Please describe it.
3. Line 288. In the description, the two types that you mention are MTC and MTE? Please mention it.
4. Line 290. Are six modern isolates that carry on MTA? Please describe it adequately.
5. Please, mention on line 294 that there are 17 modern isolates.
6. Line 298. Are seven isolates in the MTE group? Please describe it adequately.
7. Lines 306-307. Please, fix the title and caption of figure 3.
8. Line 327. Is the AB213-VUB isolate low density? Please describe it properly.
9. Line 332. Figure 4; mentions which values in cm were used to define the low, medium, and high division levels. On the height axis of the bands, mark the three levels with dotted lines.
10. Line 332-334. I suggest moving this text to the discussion and elaborating the explanation of the observation for why an isolate presents two bands within the same density gradient.
11. Line 352. Add "and Figure 4B" within the parentheses.

Discussion

1. Lines 404-406. This information is already in the results; I suggest using another text to introduce the discussion.
2. Line 417. Figure 1 should be Figure 2.
3. Line 486. Figure 1 should be Table 1, Figure 2.
4. It is mentioned that H-NS is present in the isolates analyzed in the study; however, this result is not shown in the manuscript.
5. Globally, 43 isolates of *A. baumannii* from one country are a limited group of STs; moreover, they are not representative of other geographical latitudes. *A. baumannii* has an open genome that acquires different genetic determinants over time, allowing each strain to present other evolutionary processes which may vary in each geographic region. Therefore, do you consider that your collection of modern isolates covers the full range of the existing phenotypes to propose using your platform as a reference? Thus, the proposal to consider your collection of modern isolates as a reference platform should be taken with caution and always guided by the question to be answered in each research work. My suggestion is to consider as limitations that the number of isolates is small and that it comes from a single geographical origin.

Conclusion

1. I consider that with the results of this work, it is not possible to propose a change in the design of the experiments at a global level, for which it is required that this collection of modern clinical isolates be characterized in depth, that is, that in addition to describing it at a genetic and phenotypic level, it should also be studied at a proteomic level.

Staff Comments:

Preparing Revision Guidelines

Please return the manuscript within 60 days; if you cannot complete the modification within this time period, please contact me. If you do not wish to modify the manuscript and prefer to submit it to another journal, please notify me of your decision immediately so that the manuscript may be formally withdrawn from consideration by Microbiology Spectrum.

Comments to the authors

Introduction

1. Missing references 17 and 18.

Material and methods

1. Specify which type of MLST scheme has been used.
2. In the Capsule production study indicate that the assays were performed in triplicate.
3. Reference is missing in the Virulence assay in the *Galleria mellonella* infection model.
4. There is not statistical test section. The authors do not describe the statistical analysis used to quantify bacterial cell density. Please describe it.

Results

1. Line 229. Figure 2B should be Figure 1.
2. Line 943. In the caption of figure 2 the color code and the acronyms STPas, STOx, KL and OCL are not described. Please describe it.
3. Line 288. In the description, the two types that you mention are MTC and MTE? Please mention it.
4. Line 290. Are six modern isolates that carry on MTA? Please describe it adequately.
5. Please, mention on line 294 that there are 17 modern isolates.
6. Line 298. Are seven isolates in the MTE group? Please describe it adequately.
7. Lines 306-307. Please, fix the title and caption of figure 3.
8. Line 327. Is the AB213-VUB isolate low density? Please describe it properly.
9. Line 332. Figure 4; mentions which values in cm were used to define the low, medium, and high division levels. On the height axis of the bands, mark the three levels with dotted lines.
10. Line 332-334. I suggest moving this text to the discussion and elaborating the explanation of the observation for why an isolate presents two bands within the same density gradient.
11. Line 352. Add "and Figure 4B" within the parentheses.

Discussion

1. Lines 404-406. This information is already in the results; I suggest using another text to introduce the discussion.
2. Line 417. Figure 1 should be Figure 2.
3. Line 486. Figure 1 should be Table 1, Figure 2.

4. It is mentioned that H-NS is present in the isolates analyzed in the study; however, this result is not shown in the manuscript.

5. Globally, 43 isolates of *A. baumannii* from one country are a limited group of STs; moreover, they are not representative of other geographical latitudes. *A. baumannii* has an open genome that acquires different genetic determinants over time, allowing each strain to present other evolutionary processes which may vary in each geographic region. Therefore, do you consider that your collection of modern isolates covers the full range of the existing phenotypes to propose using your platform as a reference? Thus, the proposal to consider your collection of modern isolates as a reference platform should be taken with caution and always guided by the question to be answered in each research work. My suggestion is to consider as limitations that the number of isolates is small and that it comes from a single geographical origin.

Conclusion

1. I consider that with the results of this work, it is not possible to propose a change in the design of the experiments at a global level, for which it is required that this collection of modern clinical isolates be characterized in depth, that is, that in addition to describing it at a genetic and phenotypic level, it should also be studied at a proteomic level.

Note to the editor

Dear editor,

The manuscript “Established strains of *Acinetobacter baumannii* ATCC19606, ATCC17978 and AB5075 do not reflect the whole diversity of modern clinical isolates” presents a technically sound and good quality paper. The manuscript is well written, and mastery of the topic is demonstrated. I found not significant corrections or suggestions that would limit its acceptability. Nevertheless, I would like to make the following observations in the discussion and conclusion of the paper.

The work addresses a very detailed analysis of different virulence factors of *A. baumannii* at both the genotypic and phenotypic levels. The fact that *A. baumannii* has an available genome confers characteristics that may favor its permanence in the clinic or in the environment. The authors should consider that a limitation to their work is the reduced number of isolates and that it comes from a single geographic origin.

I consider that with the results of this work, the authors cannot propose a change in the design of these experiments at a global level. To do so, they need to study a collection with more modern clinical

isolates obtained from different geographical areas. Likewise, I consider it essential that besides being characterized at the genetic and phenotypic level, they should also be described at the proteomic level. Therefore, this conclusion should be reviewed by the authors.

Brussels, the 10th of November 2022

Prof. Charles Van der Henst, PhD
VIB-VUB Group leader
Microbial Resistance and Drug Discovery
Flanders Institute for Biotechnology
Vrije Universiteit Brussel, Center for Structural Biology
Pleinlaan 2, Building E-3, 1050 Brussels (BE)
charles.vanderhenst@vub.vib.be
Office: +32 (0) 2 629 19 40

Object: Author's response to reviewer's comments (Reference: Spectrum03061-22).

New title of the manuscript: "**Phenotypic characterization and heterogeneity amongst modern clinical isolates of *Acinetobacter baumannii***"

Dear Editor,
Dear Reviewers,

We wanted to acknowledge you for your time and consideration invested in this reviewing process. We do witness that our revised manuscript is indeed an improved version. As you will notice, all the points raised by the reviewers were addressed.

Please find our point-to-point response:

Reviewer #1

General comments:

In the manuscript entitled "Established strains of *Acinetobacter baumannii* ATCC19606, ATCC17978 and AB5075 do not reflect the whole diversity of modern clinical isolates", Valeck and co-authors claim to raise awareness on the use of a handful of strains of *A. baumannii* that are shared between labs or accessible through bioresource centers.

The authors should seek help for the English and proofreading of the entire article is necessary (inappropriate use of terms (ie. proof-of-concept), some words are missing (ex. lane 136/137), the definite article "the" is often missing etc...).

The authors should also select relevant references and add several references (see specific comments).

Response: We have revised the whole manuscript, more specifically concerning English editing and the references mentioned during the reviewing process. Many thanks for this comment.

The message conveyed by the title would have been relevant ten years ago, however, with the advent of massive genome sequencing of numerous clinical strains this statement is already widely accepted by *A. baumannii* researchers. The author should reconsider the main message, and thus the title, of the article. I recommend that the article be focused on the specific diversity of the Belgium strains collection regarding capsule and virulence traits as the authors already published more thorough results regarding the antibiotic resistance in AAC. So far, this manuscript is highly descriptive, and the author should exploit the genomic data to explain the phenotypes observed (link virulence phenotype to genotype, capsule production to genotype etc...).

Response: We fully agree with the review's comment. Hence, the title was changed accordingly, and the conclusions adapted to be more focus on our collection here characterized. We have altered the manuscript and commented on the potential limitations, such as geographical origin or number of examined samples.

Overall, the results are confirmatory of previous results published by other groups. It confirms that a phenotype variability is observed between clinical isolates for capsule production, in vivo virulence, and natural transformation. Based on the results of this manuscript, the reference strain AB5075 appears as a good candidate for experimental studies (capsule production, colony morphology, virulence in *Galleria*, genetic amenability by natural transformation).

Response: We agree that, amongst current widely used strains, AB5075 paved the road for *A. baumannii* research, and represents, to a certain extent a relevant model. However, we show in our study that there are limitations to its usage as well, which should be considered for future studies, one of the main messages of our manuscript. In this context, we do believe that characterizing and comparing the phenotypes of modern clinical isolates compared to historically and broadly strains is appropriate. Many thanks for this comment as well.

More specific comments to help the authors improve the quality of the manuscript.

- lane 76: modify or add more suitable reference. The reference 8 is indeed too specific to a mean of decontamination (consider a review if necessary)

Response: We have amended this reference for a review paper. Many thanks for suggesting these improvements.

- lane 78: The reference 8 does not demonstrate the variability in transformation between isolates. Change to more relevant references

Response: We have exchanged this reference for more suitable one.

- lane 94: "Chromosomally encoded LOS is a major virulence component in Gram-negative bacteria". Reference is missing at this end of this statement.

Response: The reference was added.

- lanes 105 to 109: this information is useless for the manuscript.

Response: We agree. This part was removed, increasing at the same time the clarity of our manuscript, many thanks.

- lane 78: replace in the manuscript competence by transformation (see lane 146). As written, it sounds as if natural transformation is the only horizontal gene transfer mechanism in *A. baumannii*. Be more specific.

Response: The competence was replaced by transformation when appropriate.

- In the introduction, biofilm formation in *A. baumannii* is not introduced although presented in the results section. Add a paragraph to introduce this topic.

Response: A paragraph on biofilm was added.

- lane 135: why is there a section " Short- and long-read sequencing" in the materials and methods as the results are already published?. The authors should remove this section (simply cite in the main text the corresponding reference).

Response: This part was removed, and the corresponding reference is present in our revised manuscript.

- lane 142: again, why a section " Antimicrobial resistance genotype and phenotype" in the materials and methods as the results are already published?. The authors should remove this section.

Response: We again agree, many thanks for this statement. This part was removed.

- lane 146: the authors do not test the natural competence but natural transformation (modify in the entire manuscript competence by transformation)

Response: This was modified through the whole manuscript.

- lane 161 and results section: results of the protease activity are not presented

Response: The results of the protease and hemolytic activity are presented in the Figure 2 by "ND" as "Not Detected" as they were not observed in any of examined isolates or strains.

- lane 226: replace "genotypical" by "genotypic" or "genome" comparison

Response: The word "genotypical" was replaced by "genome".

- lane 266: the result on the difficulty on detecting the bap gene is expected as it is highly variable in size (De Gregorio et al., 2015, 10.1186/s12864-015-2136-6). Moreover, a reference is missing at the end of lane 266.

Response: Indeed, many thanks. We have added this reference to the discussion and mentioned observations by De Gregorio et al.

- lane 281: this sentence is clunky, consider rewriting

Response: The sentence was rewritten.

- lane 286: the protease or hemolytic activity results are not presented, please add some examples in supplementary materials

Response: The results of the protease and hemolytic activity can be found in the Figure 2 and are mentioned in the text as asked.

- lane 288: Figure 1 is mentioned erroneously

Response: Thank you for spotting this, it was amended.

- lane 292: how reproducible is this qualitative observation? please explain in the corresponding materials and methods section

Response: Since this experiment was carried out in biological triplicates with the same results, the reproducibility is consistently high. We have added this information to the corresponding Materials and Methods section. Many thanks for this comment.

- lane 330: how is the measure performed for AB5075 given the thickness of the density band? please explain in the materials and methods

Response: Good point, thanks a lot. The distance of the center of the band from the bottom of the microtube is measured. This information was added to the corresponding Materials and Methods section.

- lane 337: please specify between brackets the corresponding isolates and corresponding result on panel C (AB213-VUB).

Response: The paragraph describing “Density gradient” results was rewritten to be clearer. Many thanks for this comment as well.

- p16, Figure 4D: why the strain *E. coli* S17 is included in the figure but not discussed in the result section? clarify this point

Response: This is again a good point. The strain of *E. coli* S17 was added as a reference strain without capsule as control. The explanation and a reference ATCC number of *E. coli* S17 was added to the corresponding Materials and Methods section.

- lane 365: specify that the result obtained in this study confirm previous work (Whiteway et al., 2022)

Response: The reference and comment that we confirm previous work were added.

- 383: The strain AB183-VUB shares the same capsule phenotype with DSM30011 that is highly virulent for *Galleria*, please comment this seeming discrepancy (Repizo et al., 2015, 10.1371/journal.pone.0138265). The DSM30011 strain should be added to the *Galleria* assays

Response: We thank the reviewer for this comment. Indeed, the virulence of the strain DSM30011 was already characterized and published, and we have added accordingly this information into our revised manuscript. In addition, this is also true for the two ATCC strains discussed in our study, meaning ATCC19606 and ATCC17978. We have thus added this information and the corresponding references concerning these three strains (DSM30011, ATCC19606 and ATCC17978) in our revised manuscript. Many thanks for pointing this out.

- lane 533: typo in ATCC19606

Response: Thank you for spotting the typo, it was corrected.

Reviewer #2

Introduction

1. Missing references 17 and 18.

Response: Thank you for this comment, the references were checked and added.

Material and methods

1. Specify which type of MLST scheme has been used.

Response: Both Pasteur and Oxford schemes were used. This information is in the cited AAC paper (<https://doi.org/10.1128/aac.00892-22>) from our group. We have also added this information to this manuscript.

2. In the Capsule production study indicate that the assays were performed in triplicate.

Response: Indeed, the capsule production assay was performed in triplicate. This information was added in the corresponding Materials and methods section. Many thanks for this statement.

3. Reference is missing in the Virulence assay in the *Galleria mellonella* infection model.

Response: The reference was added in the corresponding Materials and methods section.

4. There is not statistical test section. The authors do not describe the statistical analysis used to quantify bacterial cell density. Please describe it.

Response: The statistical analysis was added in the corresponding Materials and methods section.

Results

1. Line 229. Figure 2B should be Figure 1.

Response: Thanks a lot for noticing this, it was amended.

2. Line 943. In the caption of figure 2 the color code and the acronyms STPas, STOX, KL and OCL are not described. Please describe it.

Response: The descriptions of acronyms and the colour code were added, many thanks for pointing this out.

3. Line 288. In the description, the two types that you mention are MTC and MTE? Please mention it.

Response: We are sorry for vague description. This paragraph was substantially rephrased to increase clarity.

4. Line 290. Are six modern isolates that carry on MTA? Please describe it adequately.

Response: We have rephrased this paragraph.

5. Please, mention on line 294 that there are 17 modern isolates.

Response: We have rephrased this paragraph.

6. Line 298. Are seven isolates in the MTE group? Please describe it adequately.

Response: We have rephrased this paragraph.

7. Lines 306-307. Please, fix the title and caption of figure 3.

Response: Thank you for noticing this as well, the title was fixed.

8. Line 327. Is the AB213-VUB isolate low density? Please describe it properly.

Response: The isolate AB213-VUB is low density (high capsulation), the sentence was rewritten.

9. Line 332. Figure 4; mentions which values in cm were used to define the low, medium, and high division levels. On the height axis of the bands, mark the three levels with dotted lines.

Response: Thank you for this appropriate suggestion, we have added dotted lines and stated which part is of low, medium, or high density. Furthermore, a table (Supplementary Table 2) was added to clearly state the capsulation level of each isolate with the corresponding strain used in the study.

10. Line 332-334. I suggest moving this text to the discussion and elaborating the explanation of the observation for why an isolate presents two bands within the same density gradient.

Response: We fully agree with the reviewer 2 and we have moved the paragraph to the Discussion and elaborated on phase variation.

11. Line 352. Add "and Figure 4B" within the parentheses.

Response: Thank you for this point, we have added "and Figure 4B".

Discussion

1. Lines 404-406. This information is already in the results; I suggest using another text to introduce the discussion.

Response: This part was rewritten to avoid repetitive introduction. Thanks a lot, as this increases the clarity of the revised manuscript.

2. Line 417. Figure 1 should be Figure 2.

Response: Thank you for noticing this, we have fixed it.

3. Line 486. Figure 1 should be Table 1, Figure 2.

Response: Thank you for noticing this too, we have fixed it.

4. It is mentioned that H-NS is present in the isolates analyzed in the study; however, this result is not shown in the manuscript.

Response: Good point again. We apologize for omitting this part, we have added both methodology and result of the H-NS analysis to the text of the manuscript.

5. Globally, 43 isolates of *A. baumannii* from one country are a limited group of STs; moreover, they are not representative of other geographical latitudes. *A. baumannii* has an open genome that acquires different genetic determinants over time, allowing each strain to present other evolutionary processes which may vary in each geographic region. Therefore, do you consider that your collection of modern isolates covers the full range of the existing phenotypes to propose using your platform as a reference? Thus, the proposal to consider your collection of modern isolates as a reference platform should be taken with caution and always guided by the question to be answered in each research work. My suggestion is to consider as limitations that the number of isolates is small and that it comes from a single geographical origin.

Response: We fully agree with the reviewer 2 about the limitations of our collection (number of isolates, geography, ...). But we do not consider our collection as a "non-exhaustive" list of isolates. Rather, this is a significant strong starting point, taking into account the heterogeneity amongst *A. baumannii* isolates. These limitations were discussed in the main text in our revised manuscript. In this context, we hope to increase the number of isolates from different countries thanks to the newly launched "Acinetobase" (74) that is referred in our revised manuscript. We have amended and added this paragraph: "This will allow selecting an appropriate strain rationally, facilitated by the new Acinetobase (74) platform, which suits the needs of ongoing studies with a particular biological question and will help to overcome bias caused by geographical origin of particular strains, will provide geno- and pheno-typic characterizations and the strains of *Acinetobacter* themselves."

Conclusion

1. I consider that with the results of this work, it is not possible to propose a change in the design of the experiments at a global level, for which it is required that this collection of modern clinical isolates be characterized in depth, that is, that in addition to describing it at a genetic and phenotypic level, it should also be studied at a proteomic level.

Response: To some extent we agree with the reviewer 2 and hence we acknowledge the limits of our study in the conclusion as follows:

"Nevertheless, we do also acknowledge limitations concerning our strain collection such as geographical bias, the number of isolates examined as well as restricting our research to specific genetic and phenotypic approaches. Further studies involving additional bacterial isolates from various geographical sites, hosts, together with microscopic imaging, proteomic, and expression analyses will be required".

Thanks to the reviewer's comments and the whole submission process, we do witness that this revised version is significantly improved. We therefore wanted to thank the reviewers for their time, and the useful and constructive feedbacks.

Sincerely

Prof. Charles Van der Henst, PhD

November 22, 2022

Prof. Charles Van der Henst
Vrije Universiteit Brussel
VIB-VUB Center for Structural Biology
Pleinlaan, 2
Brussel 1050
Belgium

Re: Spectrum03061-22R1 (Phenotypic characterization and heterogeneity amongst modern clinical isolates of *Acinetobacter baumannii*)

Dear Prof. Charles Van der Henst:

Your manuscript has been accepted, and I am forwarding it to the ASM Journals Department for publication. You will be notified when your proofs are ready to be viewed.

Sincerely,

Ayush Kumar
Editor, Microbiology Spectrum
